# The Agency of Difference in Chilean School Policies and Practices: A BioSocioCultural Way-Out Perspective

Claudia Matus * and Valentina Riberi 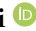

Centro Justicia Educacional, Pontificia Universidad Católica de Chile, Santiago 8331150, Chile;
valentina.riberi@uc.cl
* Correspondence: cmatusc@uc.cl

**Abstract:** In this paper, we explore the active production of difference (as lacking) through the School Vulnerability Index and the School Inclusion Law in Chile. Through a diffractive reading, we present the contradiction between these two policies. While discriminatory knowledge about school subjects is produced in the School Vulnerability Index as truth and common knowledge for the school community, the School Inclusion Law is designed to solve practices of discrimination at school. We contend that, to address issues of segregation in school settings, we have to question the kind of knowledge we need for a more democratic and just future. As a result, we trouble the separation of biological, social, and cultural realms on which instruments are based to continue segregation practices as a natural way to frame inclusion policies in educational contexts. We argue that both policies and instruments play a decisive role in the continuity of a culture of segregation in a neoliberal school tradition.

**Keywords:** diffractive reading; difference; policy instruments; state-funded discriminatory policies; segregationist public-school practices





## 1. Introduction

As common sense tells us, segregation happens, but how do segregation cultures persist despite all the critical thinking and research produced over time? As we write this paper, a major cultural and social outbreak has occurred in Chile and an unexpected worldwide pandemic is in progress. The massive protests and population unrest that arose in October 2019 are political, affective reactions to neoliberal policies and their consequent dehumanized cultural systems. As these structures of inequalities are more evident in the privatized operations of the health system, inequitable retirement pensions, and exclusionary education system, we contend that schools are critical institutions in which segregation is produced as a natural state of being.

The Chilean school system is a paradigmatic case, given its high degree of privatization and deregulation (Bellei 2015; Carrasco et al. 2019; Carrasco and Gunter 2019; Verger et al. 2016, 2017; Villalobos and Quaresma 2015) as well as its strong orientation towards competition (Falabella 2016). This, in addition to residential segregation (PNUD 2017; Santos and Elacqua 2016), makes Chile one of the most inequitable systems in the world (OECD 2019). School and residential segregation have become a naturalized way of producing cultural and social positions for subjects and communities.

As a way to address issues of segregation and exclusion, there has been a proliferation of policies to respond to different demands in Chilean schools. The School Vulnerability Index (2007), the School Inclusion Law (2015), and the School Community Law (2011) are a few of the regulations that make up this policy landscape. These regulations are designed as significant efforts to undo the effects of privatization practices of schooling driven by the neoliberal agenda in Chile. In this article, we focus on two of them, the School Inclusion Law (SIL) and the School Vulnerability Index (SVI). Even though they

are part of the initiatives to advance the notion of public education, we argue that both regulations, as policies and instruments, play a decisive role in the continuity of a culture of segregation in a neoliberal school tradition. While the School Inclusion Law promotes ideas of "students mixing" to advance the values of public education, the School Vulnerability Index consolidates systems of differentiation for students based on questionable biological, social, and cultural dispositions to school success. This takes us to a deep interrogation of the kinds of contradictory knowledges policies and instruments produce when their purpose is to transform and change schools to be just places.

Given this scenario, research on segregation and exclusionary practices produced by market-based policies in Chile is an important field of interest. Studies include documentation and descriptions of those institutional and administrative processes that took place under a major educational reform, such as free parental choice and decentralized admission systems (Carrasco et al. 2017; Contreras et al. 2010; Gutiérrez and Carrasco 2021; Hernández and Carrasco 2020; Zancajo 2019); processes of deprivatization and desegregation (Bellei 2016; Valenzuela et al. 2013; Verger et al. 2017); and the relation between educational privatization processes and broader neoliberal and neoconservative ideologies (Falabella 2016; Matus 2015, 2019; Matus et al. 2018). These critical studies have provided a rich field of discussion about the effects of educational privatization models and the advancement of cultures of inequalities.

Despite these important advances, we contend that we need to be more suspicious about the vitality of the multiplicity of processes that produce differentiation and segregation. How is it that, despite all the research available, we have not been able to achieve equality? This is the question that triggers our research interest. In order to answer this question, we turn our gaze to how state instruments and policies, as entangled identities, produce knowledge that consistently reproduces segregation. While it is true that some research studies explore the effects of instruments in the production of data, it is important to note that they do it in reference to the instrument itself (Bellei and Muñoz 2021; Meckes and Mena 2021; Campos-Martínez and Guerrero 2016; Flórez 2015), which is not enough. Understanding how instruments produce biased data is not part of our research concerns. Instead, we study how the entanglements of apparently non-related documents and practices actively reproduce segregation through the use of an essentialized difference. In this article, both a law and a measuring instrument are constitutive of the phenomenon; they are not separated elements we bring together to explain. Taking a diffractive reading as methodology, we focus on how instruments reproduce ideas of difference that later become common sense for those policies in charge of transforming schools into non-discriminatory places.

Our guiding questions are: How do segregation cultures persist? What is the responsibility of measuring instruments and policies in producing inequalities when they supposedly advance ideas of public and just education? How do cultures of segregation maintain themselves based on a distinctive and self-explanatory separation of the biological, social, and cultural realms promoted in measuring instruments?

In what follows, we present our theoretical and methodological frameworks to position our analysis and discussion. Then, we present pieces of the ethnographic work produced in three schools in Chile during 2017, in which our purpose was to document the contradictions these participating schools were experiencing when implementing the School Inclusion Law. Along with this, we provide a description of the law itself and its purposes, to contextualize the fieldwork. Later, we describe and critically analyze the School Vulnerability Index as an active source that produces knowledge about school subjects as vulnerable. We contend that this measuring device reproduces a stability of categories that is problematic when we question the persistence of inequalities in the context of a major public-school reform. As a result, a diffractive reading of both, apparently, disconnected pieces of information, shows the workings of instruments as producers of essentialized difference, and how this difference is lived in daily school interactions. We present an argument on how the separation of the biological, social, and cultural realms

used to define and produce vulnerability in the instruments is political, since it produces a kind of knowledge that is partial and, therefore, reproduces stigmatized ideas of humans, as we show in the implementation of the Inclusion Law.

## 2. Theoretical and Methodological Perspectives

In order to pursue our analysis, we position ourselves in a performative posthumanism perspective (Barad 1999, 2003, 2007; Bennett 2010; Haraway 2016; Tsing 2015) to engage in the complexity surrounding the entangled production of the segregation phenomenon in school practices of knowledge and the instruments that separate and give different statuses to the biological, social, and cultural realms used to produce vulnerability in schools. This perspective provides us with important points for analytical work such as: (a) a decentering of the human; (b) the recognition of the liveliness of matter; (c) nomadic thought; (d) the problematizing of binaries; (e) the notion that measurement instruments are part of the ontology; and, finally, (f) the consideration of interconnectedness (assemblages, entanglements) rather than a division between macro- and micro-level phenomena (Gullion 2018, p. 23). In what follows, we focus on two particular elements of this list in an effort to illuminate our analysis. First, we problematize the normative power of binaries and the unquestioned stability of categories presented, particularly, in the School Vulnerability Index, in which biological, social, and cultural dominant discourses work together as justifications to monitor the present and to anticipate the future of school subjects. While biological discourses reinforce the idea of the proper body and brain, the social dimension presented in this instrument insists on the benefit of particular dispositions to organize institutional life (e.g., heterosexual family) whereas dominant cultural discourses (e.g., showing interest in books or for reading) promise a moral present and future for students and communities. Second, we contend that instruments, as part of the ontology of the phenomena, change people's ways of being in relation to what instruments produce as knowledge. In the case of the School Vulnerability Index, it is an active producer of the continuation of paths of inequality by assuring truth regarding vulnerability. We state that this is made through the simplification and reification of statuses between the biological, social, and cultural realms used to define vulnerable school subjects. In this case, the index "produce[s] correlations between 'instruments' and ['subjects']" (Barad 2007, p. 344) that later become characteristics and attributes students 'possess' and 'exhibit' and that are 'easy' to recognize. Therefore, vulnerability becomes a reality independent of the instrument itself. Therefore, reality is imagined as a stable picture from where to capture attributes and characteristics to describe school subjects as real templates or reflections of that reality. In this framework, the dynamic and contingent mutual materializations of divergent and unpredictable biological, social, and cultural processes are out of consideration.

As we focus on the operations of this instrument in producing vulnerability as a menace and marginality, we argue that these actions work against the public education agenda promoted by the School Inclusion Law. To consider the policy and instrument diffractively means that they are both dimensions of the same phenomena; they are not separate entities or technologies that we analyze separately and then put together. In other words, to read the policy and instrument diffractively does not mean that we force them together to explain; instead, we read them together as dimensions of the same phenomena, which allows for any unforeseen aspects to emerge as a critical aspect to change the nature of the phenomena.

A diffractive reading proposes a non-representational approach to deconstruct the usual way of reading texts (Barad 2003, 2007, 2010, 2014; Haraway 2016; Juelskjær et al. 2021; Murris and Bozalek 2019; Schrader 2012). A diffractive reading is different from other critiques of texts in that texts are relationally read through each other, looking for and strengthening unexpected provocations and possibilities to understand the phenomenon otherwise. As Schrader (2012) explains: "diffraction patterns record the history of interaction, interference, reinforcement, difference. Diffraction is about heterogeneous history, not about originals. Unlike reflections, diffractions do not displace the same elsewhere" (cited in

Juelskjær et al. 2021, p. 48). To understand how concepts such as segregation, essentialized differences, and inequalities bend and spread, move and combine to become segregation, we establish essentialized difference and inequality as part of our research agenda.

Methodologically, in order to engage in a diffractive reading, both the ethnographic data produced in the study of the implementation of the School Inclusion Law and the instrument used to measure vulnerability are considered empirical materials (Denzin 2013), as they are interlocking moments of the phenomenon of segregation school practices.

Thus, the materials used for this analysis were (1) ethnographic data of a study whose purpose was to analyze the effects of the implementation of the School Inclusion Law in school practices, in particular those that referred to the principle of non-discrimination and inclusion proposed in the law. This study was conducted in three schools in Chile in 2017. In this ethnography, we paid particular attention to the intersections of gender, race, sexuality, social class, age, and ability. (2) The second material is the SVI and its statistics ranging from 2010 to 2019. The SVI statistics are public and available on the JUNAEB webpage. Data are disaggregated at the school level, where only the number of students by vulnerability priority is public. We worked with these datasets to explore how the school vulnerability index evolved in a decade and which students (socioeconomic status, rurality, regions, and school dependency—public or private school) are defined as vulnerable. We also included the decision tree that was used to estimate the SVI as material.

To read both materials as part of the same phenomenon requires that we think of them as in a "flat ontology" (Barad 2007), which means that neither of them explain the other; they belong to the same ontological status. In other words, we move away from the causal relation between policies and implementations, instruments and impact, or macro- and microanalyses. To think of these two materials as constitutive of the phenomenon and to question how they interfere with each other gives us a rich opportunity for analytical inquiry. It is important to note that our unit of analysis is not people or school communities, but the phenomena of segregation itself.

To produce those iterative patterns that explain the entangled relation of these two materials in the production of segregation, we read them using the following questions: What are the binary orders that transverse both materials? What are the material and discursive dimensions that justify segregation and difference as lacking? Where do the unfoldings of these questions take us to understand issues of injustice, segregation, and difference as lacking? How do these two pieces, together, tell us something about the persistence of common sense to explain segregation in schools? How can we trace material–discursive practices to explain and frame the problem otherwise?

To answer these questions, we provide a description of the two interlocking moments of the analysis: the School Inclusion Law and the School Vulnerability Index. In addition to providing a critical description of each of the texts (law and instrument), we highlight different themes to exemplify the workings of particular material–discursive relations that justify difference as a vital component of segregation school practices.

## 3. Analysis

### 3.1. The School Inclusion Law

As part of a one-year ethnographic study in three schools in two different regions in Chile, we documented in-place practices produced by the implementation of the School Inclusion Law (Ley N°20.845). This law is an effort by the Chilean government to improve equity and quality in education. As of March 2016 and until 2019, all municipal and private-subsidized schools were required to implement this law, which is based on three main principles: (1) the end of profit; (2) the end of co-payment in educational systems financed with public resources, and (3) non-arbitrary discrimination and inclusion. Specifically, the School Inclusion Law prohibits the selection of students when entering school institutions as well as additional school fees when it comes to establishments that are financed with public resources. This law only applies to public and private-subsidized establishments (92% of the total number of schools in Chile), leaving out the remaining 8% of establishments

represented by the private school sector. Considering the culture of segregation in Chile, the implementation of this reform has intensified practices of student identification (of subjects, groups, families, establishments, etc.) from which new or other forms of exclusion and unequal distribution of learning are produced (Matus et al. 2018). As previous research shows, when school communities think of themselves as more heterogeneous due to the non-selection of students, they install ideas about integrated students as individuals who do not represent the values or customs of the establishment or as students who do not have a recognizable brand. This is translated into phrases such as 'the other families and communities', or 'the other types of children' that make up the school community. In other words, one of the problems is that this law is being implemented in a highly segregated context that does not consider the entrenched common-sense ideas around homogeneity and normalizing practices of essentialized difference, or even how discriminatory institutional policies operate (Matus and Haye 2015; Matus and Rojas 2015).

These essentialized practices of knowledge around students who are not selected to enter schools are part of reactions, particularly from parents and guardians who are worried about their children being mixed with 'other types of kids in schools'. This might be understood as how neoliberal agendas have engrained conservative ideas of high- and low-status cultures as separated and that only now (because of the implementation of this law) are mixed in school contexts. The similar vision in-service teachers have about not-selected students is also particularly interesting. They note that: 'There are going to be bigger problems regarding the lack of discipline'; 'We have had to hire new professionals who have not been historically linked to the field of education, for example, lawyers, and we have also had to increase the hiring of educational psychologists'; 'We are going to need more moral, social, [and] psychological evaluations, and [we need] to assess risk'.

In what follows, we present excerpts from the participants of schools where we highlight these critical practices of knowledge used to segregate students based on ideas of vulnerability.

The Assurance of Normalcy through Selection

In our first meeting with the principal in School 1, she explained to us that one of the difficulties they faced when they decided to adhere to the School Inclusion Law was the concern of the parents and guardians when they found out that the school would be transformed into a tuition-free school. According to the principal, for parents and guardians, the word *free* is the same as the word *municipal* (no one pays for education, which is at the center of the public education reform). For the principal, the word *municipal* translated into *insecurity* for both teachers and parents. In other words, "if you pay for school you are safe—everybody is the same" (the premise behind this is that, since they used to have the right to select students, vulnerable ones were not allowed in the community). In this case, the association of paying for education allows parents to imagine that all the students will be similar, which prevents their own children from socializing with these 'other students' who share different values.

On the other hand, the subsidized school owner says he understands the concern of parents and guardians because he considers that: 'parents and guardians believed that families who are not very committed would enroll [at the school] because there is also an issue of family commitment'. According to the school owner, parents and guardians prefer a private-subsidized school due to its security. In relation to this, he explains: 'it has been proven that a private-subsidized school is safer than a municipal school, whether we like it or not' (School Owner Interview, Establishment 1).

In another participating school, which is a high-performance institution, teachers describe their students as highly committed to the academic demand, and they explain that this is because they were selected. Now that they cannot select students they are afraid of losing their institutional prestige. When talking about new students, they say, *'we don't want fleabags'*, *'trashy'*, or *'vulnerable students'*, and they state that the new students could be a bad influence on the students that are already enrolled in the school.

To make this problem more explicit, in a conversation with a couple of teachers, they commented that the issue of non-selection had been complicated since it was first brought to the table. The main fear was the arrival of other types of children. The ethnographer asked what children they refer to, and the teachers answer that they refer to children who are 100% vulnerable and a little mischievous, but that they value the preparation that the school did to receive 'these children'.

'It has been difficult to set a standard for children and parents and guardians for a while because vulnerable children already know the law backwards and forwards, so they know that they cannot be expelled from the establishments'. She (a teacher in school 2) then continued, 'what happens is that the old parents and guardians feel that the school [by selecting students] gives them the assurance that their child is going to leave [the school] with manners or that he is going to leave [the school] with proper hair, that he is not going to wear earrings, and a host of other things. So, if the school allows a [male] student with an earring to enter, [or] to put on makeup, or to have a football player hairstyle, we would be allowing a quote unquote trashy person to enter the school. So, letting in a kid with a shaved head or a kid with unkempt hair means that bad influence enters the school. It means that these new students are going to infect this little boy who was orderly, you see? I believe that this is one of the problems within the process of inclusion in the school. In fact, parents and guardians asked us about the new students: Are they going to come dressed any way they want? Are they going to do what they want? Are they going to cuss all day?' (Teacher, School 2, Fieldnotes, 2017).

Now, where do all these conceptualizations about 'other students' and 'other communities' come from? And most importantly, where is this knowledge produced? In what follows, we present the School Vulnerability Index as a particular moment in the production of cultures of essentialized difference. We contend that the ways school agents talk about 'the other' as a menace can be traced back to the ways different policies, that intend to advance issues of non-discrimination and inclusion, produce knowledge and appraisals of subjects and communities.

### 3.2. The School Vulnerability Index (SVI)

The SVI is an instrument that has been used since 2007 to measure and assign vulnerability levels to every student attending public and private-subsidized Chilean schools. It was designed and implemented by the National School and Scholarship Assistance Council (JUNAEB in its Spanish acronym), and it defines vulnerability as:

"The dynamic condition that results from the interaction of a multiplicity of risk factors and individual and contextual protectors (family-school-neighborhood-municipality) before and during the educational development of a child, which is manifested in behaviors or events of greater or lower social, economic, psychological, cultural, environmental, and/or biological risk, producing a comparative disadvantage between subjects, families, and/or communities" (JUNAEB 2005, p. 48).

It is important to note that this index is intended for the public and subsidized school sectors, which comprises 92% of the total school population. The 8% corresponding to private schooling is not part of this policy to measure and produce vulnerability. In other words, vulnerability as a biological, social, and cultural prescription used to produce the category and label of vulnerable schools and students assumes, from the very beginning, that 8% of the Chilean population (that corresponds to the private elite section of the school system) do not feature any of the dimensions from which vulnerability is produced.

To estimate school vulnerability, JUNAEB uses administrative data and a parent survey. This survey is distributed in paper form to students in kindergarten, 1st, 5th, and 9th grades in public and private-subsidized schools in the country. The survey is arranged into six groups of questions. The first group aims to identify students (name, ethnicity, nationality, residency, among other basic information). A second group collects information about family composition and organization (e.g., number of people who live in the household, head of the household, parents' school levels and current occupations, child sleeping habits,

existence of a special place for the child to study, if the home is near recreational places or hospitals, and, finally, the person who takes care of and does homework with the child). Another group of questions is oriented toward finding out about the student's health (e.g., detailing problems on cognitive tasks, visual reports, learning or control behavioral problems, or any diagnosis of chronic illness). Fourth, the survey presents a set of questions regarding aspects of early childhood upbringing (e.g., mother's age at the child's birth, the child's birth weight, if he/she was premature, the type of lactation received, the type of schooling, early childhood stimulation, and, finally, if the father was physically or economically present during the child's upbringing). The next group of questions inquires about family context and social relations (if someone in the family has been in jail; if the child works; if the father, mother, or guardian reads; and participation in social organizations). Finally, there is a group of questions concerning the lifestyles, characteristics, and expectations of the student. Some of the questions presented in this section are related to if the child shows affection to parents and family; expresses emotions and feelings to other people; shows emotions and feelings through bodily expressions (cuddling, hugging); plays with other boys and girls; shares her/his belongings; is explosive or aggressive with people; participates actively in physical activities; asks adults questions to understand or clarify ideas; shows interest in reading and books; shows interest in understanding the environment and context that she/he is in; and assembles and disassembles things as a form of play. Also, parents are asked about their expectations about the student's school performance.

As presented, vulnerability produced by this instrument becomes a particular appraisal of biological, social, and cultural dispositions. In other words, vulnerable subjects are produced as those diverging from an ideal biological, social, and cultural norm, in which 'the norm remains invisible and uncontested' (Koivunen et al. 2018, p. 5). This index itself speaks from a norm and as such, it also reinforces ideas about what is not vulnerable, namely, white, heterosexual, patriarchal, bourgeois, abled subjects.

As this instrument frames the endurable idea that vulnerable students are biologically, socially, and culturally deficient or lacking, racist, classist, and gendered systems become a commonsense practice of knowledge. As Hintz et al. (2018) note, 'This is ultimately a story of data justice, whereby the unquestioned acceptance of numerical 'facts' can further perpetuate structural inequalities and injustice' (cited in McCallum et al. 2020, p. 2). As we showed above, the ways teachers and administrative personnel talk about students that have to be admitted because of the non-selection criteria describe the active and political intertwining processes between knowledge production (the instrument) and practices of knowledge (schools implementing the law) to produce 'the reality of vulnerable school subjects'.

In what follows, we highlight three elements worth considering from this instrument for our analysis.

### 3.2.1. Non-Traditional Motherhood as a Predictor of Vulnerability

Motherhood appears as a critical practice to define the value of a cultural practice and to reaffirm the social institution of heterosexuality. Current mothers, mothers who breastfeed, mothers whose first pregnancy was not during adolescence but rather at the 'right time', and mothers that were not alone when the child was born are highly valued characteristics to define a vulnerable school student. This normative idea of motherhood, as desirable and connected to vulnerability for children and young people, is a way to causally connect social institutions (heterosexuality, normative femininity), and cultural dispositions (the practice of the right motherhood) to biological conditions (e.g., breastfeeding, baby weight) that might explain behavior (Karpin and O'Connell 2015, p. 1482). This is where biological explanations appear to hold a more 'objective' and causal explanation for those cultural practices that are understood as school success barriers.

Therefore, this frame serves to justify the powerful and problematic way of reasoning the so-called 'natural' conditions a student is born into, as it sets a limit for cultural potentialities to explain school performances.

### 3.2.2. Number of Tooth Cavities

An interesting question the SVI includes on its survey and algorithmic formula is the number of cavities that a child had before the age of 5. As stated, it seems that cavities occur as a purely cultural phenomenon: they happen to be a problem in those people with poor personal hygiene habits and those who follow a poor diet. As it is, children with more cavities are described as vulnerable, which connects the problem of cavities to a cultural practice. Instead, we understand cavities as more than a cultural phenomenon; instead, they are where a multiplicity of sociodemographic, dietary, breastfeeding, oral hygiene, bacteria flora, environmental (the use of rain or well water as drinking water) factors, among others (Kirthiga et al. 2019; Peltzer and Mongkolchati 2015), are potential reasons and explanations for cavities. Even more, these factors, as they differ between income levels, are more present in high- or upper-middle-income levels countries where the frequent consumption of sweetened foods, poor oral hygiene, and the presence of visible plaque are the strongest risk factors associated with early childhood cavities (Kirthiga et al. 2019). Therefore, these risk factors are not only present in 'vulnerable communities'.

Thus, cavities reflect a strategic use of culture to blame students and to justify social inequality. At the same time, cavities as a distinctive reality of vulnerable students neutralize the structural segregation they reproduce, in which the 8% that is not measured by this instrument (those students belonging to the private sector) can opt for expensive dental services from an early age even with poor oral hygiene and diet.

### 3.2.3. Lifestyles

The SVI instrument interestingly uses particular 'lifestyles' as indicators of disadvantage[1]. The given descriptions of 'lifestyles' in the instrument refer to students showing their feelings through bodily expressions (hugs, holding, etc.), getting together to talk and hang out with friends, posing questions to adults to clarify certain topics, showing an interest in books and reading, and so on. These descriptions of desirable social and cultural practices are presented or imagined as 'authentic' marks of particular identities that a sociable citizen is expected to have in a neoliberal culture. As Ellen Samuels (2014) notes '[...] fantasies of identification have never really been about science. They are about culture, about politics, about the rule of law and the unruliness of bodies,' (p. 186) which means that to connect these questionable lifestyles to vulnerability implies that being affectionate and sociable is a desirable attribute for successful students.

These three factors (motherhood, tooth cavities, and lifestyles) allow us to think that the School Vulnerability Index imagines culture as a recognizable influence on those natural conditions a student is born into, and, at the same time, that nature sets a limit for cultural and social potentialities (Duster 2003; Gillborn 2008, 2010, 2016; Roberts 2012; Riberi et al. 2021; Rosiek and Kinslow 2016). This way of thinking frames particular conditions to explain differences in school achievement, provides reasons to normalize behaviors, justifies discourses of intervention, and promotes neoconservative agendas about families and children (Matus 2015). We argue that the active production of systems of identification, differentiation, and labeling in Chilean policy documents requires a critical examination, since they not only use particular characteristics and attributes of subjects to define what are deemed to be the 'risk factors', but they also value and legitimize conservative ways of organizing intimate and public lives.

The ways in which the SVI uncritically reproduces a western, bourgeois, heterosexual, middle-class culture invite us to question not only notions of naturalized cultures, but also the dynamic and relational notion of the biological, the social, and the cultural within the imagined stabilizing condition of measurement practices.

## 4. Conclusions: A BioSocioCultural Perspective as a Possibility to Transform Knowledge

A diffractive reading of both the law and the instrument as dimensions of the same phenomena focuses our attention on how they are intertwined to produce a phenomenon of difference as vulnerability, and how this knowledge bends and spreads, moves and combines, taking different shapes. As we have shown in this article, thinking about segregation as something that *happens* in schools as separate from the instrument that produces vulnerability and the law that promotes "mixing" hides the productive forces this particular knowledge has in the persistence of inequalities. As part of our analysis, we have come to understand that, for vulnerable school subjects to be real and intelligible, particular actions need to occur: first, the instrument (SVI) must actively separate biological, social, and cultural realms to strategically provide causal and linear explanations for risky human behaviors; second, school practices of knowledge (as the ones presented in the ethnographic narratives presenting the implementation of the SIL) must be critical of the knowledge produced in instruments; and third, the products of both, the SVI and the SIL, need to be understood as separate entities: one measures vulnerability and the other solves issues of discrimination.

This diffractive reading allowed us to critically rethink the separation with which the SVI understands and gives life to the biological, social, and cultural worlds as well as questions those hierarchical and causal justifications we grant one realm over the other to explain who we are and who we will become. As we showed, biological dispositions are seen as the basis for good school performance, while belonging to recognizable social institutions (e.g., traditional heterosexual family) facilitates personal progress, just as valuable cultural practices anticipate school success; we claim that, in order for these statements to operate as truths, they need to be reasoned as separated realms. Constructing human experiences as segmented and as one explaining the other, as this instrument does, allows the persistence of unfair reasons for how 'certain kinds of school subjects' do not succeed. This exclusionary knowledge becomes common sense in schools to understand human experiences.

For instance, we wonder how the SVI will answer the following questions: What notions of 'the biological' does the instrument hold on to? What ideas of 'the social' does the instrument privilege? How does the instrument think about 'the cultural?' More importantly, where does the instrument locate those lines that separate these three dimensions? What are the effects of this separation when addressing issues of inequalities? To interpolate the instrument with these questions allows us to show the critical effects of an instrument rooted in the idealization of an abstract model of the biological, social, and cultural—as separated and hierarchically organized systems—and the dangerous production of unproblematized 'kinds' of subjects to be reasoned as vulnerable (or not).

On the one hand, we have shown how the SVI stabilizes sexism, heterosexuality, patriarchy, bourgeois, elite, and whiteness as property, as unquestionable orders from where to locate subjects, discourses, and practices. On the other, the SIL aligns discourses of openness, public as good, and non-discriminatory practices as the solution to address exclusionary practices. In other words, while one produces essentialized, unescapable difference, the other promotes mixing to solve exclusionary practices that are the effects of the production of essentialized difference. When these two happenings clash as they do, we have to move beyond language and discourse and engage in the materiality of the production of difference (instruments and laws) because they reveal aspects of an empirical reality that depend entirely on them. When we look at them together, working together, we are interested in the ways they work and how the boundaries of binaries are shaped and reshaped. What we have done is to understand how interferences between the biological, social, and cultural altogether presented at different intensities are important without putting them in opposition. We have explored the tracing of concepts that produce and solidify the notion of difference as lacking; and how they fold, unfold, and change. By doing this, we have shown how difference is made, and what the effect of this difference is.

To transform the persistent inequalities we live in, we need to question what measuring instruments do, and what the life trajectories they make real for school subjects are. As the School Vulnerability Index reinforces school segregation and produces vulnerability in relation to bodily, social, spatial, cultural, and affective differentiations of a fictional norm (Davis 2014), it activates normative practices supported by imaginations of school subjects as well-defined and static identities; social realms as where healthy relations happen; and monolithic ideas of culture are seen as distinctive dynamic influences that differentiate one community or group from another. Instead, we contend that the biological, the social, and the cultural 'are co-constitutive, entangled, and variable cuts from the same larger fabric and point to new questions and possibilities [to study the persistence of inequalities]' (Dixon-Román 2017, p. 13). If not seen this way, dominant notions of the biological, the social, and the cultural, as separate, distinctive, and independent dimensions, continue to act as forces that produce inequalities from where to explain different human capabilities and possible futures.

We argue that, to address issues of sexism, racism, classism, and several other systems of differentiation that have become part of the common way to practice school knowledge, as we present in the ethnographic passages, we need to question the separation of the biological, social, and cultural realms as natural and intelligible ways to frame the world, and realize that, as such, this separation acts as a foundation for the constitution of binaries that later translate into, for instance, gender essentialisms and the prevalence of racist taxonomies (Alaimo 2016; Barad 2007, 2011, 2012; Frost 2016; Willey 2016).

As we move forward in our thinking, we wonder what kind of knowledge about ourselves might be produced if we think about the biological, social, and cultural realms with no causal relations among them. To think about the biological, social, and cultural operating at the same level with no status or one coming before the other may help us address the vitality of life when producing knowledge to inform policies (Matus 2019). This is what we call a BioSocioCultural perspective.[2] This perspective entangles the potentialities of these three dimensions that give life to the explanations of who we are and, in relation to a BioSocioCultural frame, to think about the object of study, to inform an instrument to measure lives, and to provide a framework to complexify our analysis may put us on the path to a different future. A BioSocioCultural perspective is an intertwined epistemological, ontological, and affective way of thinking and knowing, where none of the three are precedent or an origin to explain or justify the other. This provides a way to capture the vitality of life that exceeds our humanist ways of noticing (Tsing 2015) and directs our attention to life as constantly evolving, moving, and becoming. We need a vital and unlimited *now* to intraconnect all these realms to provide different explanations about the meanings of life we give others, which finally define their and our life trajectories. If instruments—as producers of knowledge about vulnerable school subjects—and the consequent practice of this knowledge in schools, are not seen as in a critical relation to continue those irreversible paths of inequalities, then we must make ourselves responsible for defining unequal and precarious futures for those 92% of the Chilean school population who are the target of the policies presented in this article. Can we think of this notion of vulnerability produced in the SVI as generating social change or only more vulnerability? We contend that the notion of vulnerability produced in this instrument is supported by other instruments and state practices that, as a whole, support various political agendas, including classist, paternalistic, and misogynist ones. Therefore, the elimination of this instrument will not change the fact that vulnerability is thought of as marginalization. We argue that, since the SVI perpetuates the idea of vulnerability as marginalization due to how groups are categorized, we need to reconsider how this instrument continues the endless circle of segregation. When vulnerability is produced as an intrinsic quality of a group or people exhibiting less merit than others, then we have a problem to address beyond the instrument itself. If the SVI is actively producing " . . . those that embody difference from the normative subject" (Koivunen et al. 2018, p. 12) and these ideas are easily appropriated by dominant groups, then we need to question what it is used for.

Examining the impact of how the processes of naming and framing people or groups as vulnerable is highly political.

For education policies and associated instruments to engage with social justice in ways that do not reinforce existing inequalities, it is critical to discuss the hidden biological, social, and cultural arrangements used to explain and justify the production of vulnerable school subjects.

**Author Contributions:** Conceptualization, C.M.; methodology, C.M. and V.R.; formal analysis, C.M. and V.R.; investigation, C.M. and V.R.; writing—original draft preparation, C.M.; writing—review and editing, C.M. and V.R.; supervision, C.M.; project administration, C.M.; funding acquisition, C.M. All authors have read and agreed to the published version of the manuscript.

**Funding:** This research was funded by the Chilean National Agency for Research and Development (Agencia Nacional de Investigación y Desarrollo de Chile, ANID), CIE160007, PIA SOC180023, and FONDECYT 1160732.

**Institutional Review Board Statement:** Not applicable.

**Informed Consent Statement:** Informed consent was obtained from all subjects involved in the study.

**Data Availability Statement:** Not applicable.

**Acknowledgments:** The Center for Advanced Studies in Educational Justice (https://centrojusticia educacional.uc.cl/center-educational-justice/) and the Interdisciplinary Research Platform Normalcy, Difference and Education (www.nde.cl).

**Conflicts of Interest:** The authors declare no conflict of interest.

## Notes

[1]   For writing purposes, we will refer to the concepts of disadvantage and vulnerability indistinctly.

[2]   The corresponding author created the research line BioSocioCultural Inclusion: Challenging Homogeneity in Contemporary School Settings at the Center for Advanced Studies in Educational Justice in Chile. The second author is part of the research team. During the last five years, this research group has been producing a theoretical field of discussion and rehearsing research practices to advance a BioSocioCultural perspective to address issues of inequalities.

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
