# Peer review of "The Agency of Difference in Chilean School Policies and Practices: A BioSocioCultural Way-Out Perspective"

_socsci, doi:10.3390/socsci11070313_

Round 1

Reviewer 1 Report

This is a very interesting and thought provoking article which I quite enjoyed reading.

My suggestions for change are relatively minor, and I have provided a bit more detail on the attached file to help.

The main changes I think are important are 1) the need to include a methodology, 2) the need for a conclusion, and 3) the separation of the description of the Law, from the findings.

I found the analysis based on the biological/social/cultural justifiable, though to me the findings presented and the subsequent analysis talk more about values (axology) and identities (ontology) which are embedded in both documents discussed. There is obviously a clash of values which lead to confused messaging about identities for students and families in the schools in question. At the end, I wonder if you could offer a way forward? What would happen if you abolished the SVI? Would that make a difference?

Reviewer 2 Report

It was a great pleasure reviewing this paper. The paper is well written and it significantly contributes to the field of educational policy. The authors are using diffractive reading in order to analyze the structure and purpose of the School Vulnerability Index in the context of the implementation of the School Inclusion Law in Chile.  The paper would benefit from the citation/reference to the diffractive reading methodology.

Reviewer 3 Report

The manuscript presents a critical social debate. This social debate with a detailed description of the school inclusion law and the school vulnerability index, makes an intensive and deep dilemma in Chilean educational system. Although, the manuscript looks like a book chapter more than research article. The following comments could help in improving the manuscript:

Major comments:

-        Authors should present research gap, questions, objectives, methodology and other research elements.

-       Finding, discussion and conclusions should be created due to proposed research questions.

Minor comments

-         Authors are advise to review the following “the coronavirus as “a political actor” (Cohen 2011) has shown (lines 24-25).

-       Abbreviations should be written completely for the first time.

Round 2

Reviewer 1 Report

I appreciate the additional work done to improve the paper. In your response to my review you state: "Instead of answering this question, we think that because of what the SVI produces, it would be better to assume that most students are vulnerable." I think you need to put this premise directly into the paper, if this is the case. I think my question still stands unanswered. If there wasn't an SVI would it make a difference? Maybe none at all, but I think you are implying that the SVI produces an impression of vulnerability based on the constructs you present. Perhaps you need to make it clear that you arent arguing for removal of the SVI. And if not, what changes would you like to see?  

Author Response

Dear reviewer 1: we completely agree with the comment. We wrote a paragraph in which we answer to your question explicitly. This paragraph was added at the end of the article. 

This is the paragraph:

Can we think of this notion of vulnerability produced in the SVI as generating social change or only more vulnerability? We contend that the notion of vulnerability produced in this instrument is supported by other instruments and State practices that as a whole support various political agendas, including classist, paternalistic, and misogynist ones. Therefore, the elimination of this instrument will not change the fact that vulnerability is thought of as marginalization. We argue that since the SVI perpetuates the idea of vulnerability as marginalization due to how groups are categorized, then we need to reconsider how this instrument continues the endless circle of segregation. When vulnerability is produced as an intrinsic quality of a group or people exhibiting less merit than others then we have a problem to address beyond the instrument itself.  

If the SVI is actively producing “. . . those that embody difference from the normative subject” (Koivunen et al. 2018, p. 12) and these ideas are easily appropriated by dominant groups, then we need to question what it is used for. Examining the impact of how the processes of naming and framing people or groups as vulnerable is highly political. 

Reviewer 3 Report

Authors considered all the comments.
